# INTERPRETATIONS ARE USEFUL: PENALIZING EXPLANATIONS TO ALIGN NEURAL NETWORKS WITH PRIOR KNOWLEDGE

## ABSTRACT

For an explanation of a deep learning model to be effective, it must provide both insight into a model and suggest a corresponding action in order to achieve some objective. Too often, the litany of proposed explainable deep learning methods stop at the first step, providing practitioners with insight into a model, but no way to act on it. In this paper, we propose contextual decomposition explanation penalization (CDEP), a method which enables practitioners to leverage existing explanation methods in order to increase the predictive accuracy of deep learning models. In particular, when shown that a model has incorrectly assigned importance to some features, CDEP enables practitioners to correct these errors by directly regularizing the provided explanations. Using explanations provided by contextual decomposition (CD) (Murdoch et al., 2018), we demonstrate the ability of our method to increase performance on an array of toy and real datasets.

## 1 INTRODUCTION

In recent years, neural networks have demonstrated strong predictive performance across a wide variety of settings. However, in order to achieve that accuracy, they sometimes latch onto spurious correlations, leading to undesirable behavior as a result of dataset bias (Winkler et al., 2019), racial and ethnic stereotypes (Garg et al., 2018), or simply overfitting. While recent work into explaining neural network predictions (Murdoch et al., 2019; Doshi-Velez & Kim, 2017) has demonstrated an ability to uncover the relationships learned by a model, it is still unclear how to actually alter the model in order to remove incorrect, or undesirable, relationships.

We introduce contextual decomposition explanation penalization (CDEP), a method which leverages existing explanation techniques for neural networks in order to prevent a model from learning unwanted relationships and ultimately improve predictive accuracy.

Given particular importance scores, CDEP works by allowing the user to directly penalize importances of certain features, or interactions. This forces the neural network to not only produce the correct prediction, but also the correct explanation for that prediction. While we focus on contextual decomposition (CD) (Murdoch et al., 2018; Singh et al., 2018), which allows the penalization of both feature importances and interactions, CDEP can be readily adapted for existing interpretation techniques, as long as they are differentiable. Moreover, CDEP is a general technique, which can be applied to arbitrary neural network architectures, and is orders of magnitude faster and more memory efficient than recent gradient-based methods, allowing its use on meaningful datasets.

In order to demonstrate the effectiveness of CDEP, we conducted experiments across a wide variety of tasks. In the prediction of skin cancer from images, CDEP improves the prediction of a classifier by teaching it to ignore spurious confounding variables present in the training data. In a colored MNIST task, CDEP allows the network to focus on a digit's shape rather than its color (with no extra human annotation needed). Finally, a toy example using text classification shows how the penalization can help a network avoid a bias towards particular words, such as those involving gender.

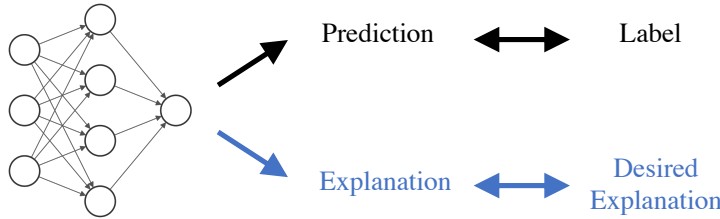

Figure 1: CDEP allows a practitioner to penalize both a model's prediction and the corresponding explanation.

## 2 BACKGROUND

**Explanation methods**   Many methods have been developed to help explain the learned relationships contained in a DNN. For local, or prediction-level, explanation, most prior work has focused on assigning importance to individual features, such as pixels in an image or words in a document. There are several methods that give feature-level importance for different architectures. They can be categorized as gradient-based (Springenberg et al., 2014; Sundararajan et al., 2017; Selvaraju et al., 2016; Baehrens et al., 2010; Rieger & Hansen, 2019), decomposition-based (Murdoch & Szlam, 2017; Shrikumar et al., 2016; Bach et al., 2015) and others (Dabkowski & Gal, 2017; Fong & Vedaldi, 2017; Ribeiro et al., 2016; Zintgraf et al., 2017), with many similarities among the methods (Ancona et al., 2018; Lundberg & Lee, 2017). However, many of these methods have thus far been poorly evaluated (Adebayo et al., 2018; Nie et al., 2018), casting doubt on their usefulness. Another line of work, which we build upon, has focused on uncovering interactions between features, in addition to feature importances, (Murdoch et al., 2018), and using those interactions to create a hierarchy of features displaying the model's prediction process (Singh et al., 2018).

**Uses of explanation methods**   While much work has been put into developing methods for explaining DNNs, relatively little work has explored the potential to use these explanations to help build a better model. Some recent work proposes forcing models to attend to regions of the input which are known to be important (Burns et al., 2018; Mitsuhara et al., 2019), although it is important to note that attention is often not the same as explanation (Jain & Wallace, 2019). An alternative line of work proposes penalizing the gradients of a neural network to match human-provided binary annotations and shows the possibility to improve performance (Ross et al., 2017) and adversarial robustness (Ross & Doshi-Velez, 2018). Two recent papers extend these ideas by penalizing attributions for natural language models (Liu & Avci, 2019) and penalizing a modified gradient-based score to produce smooth attributions (Erion et al., 2019). Predating deep learning, (Zaidan et al., 2007) consider the use of "annotator rationales" in sentiment analysis to train support vector machines. This work on annotator rationales was recently extended to show improved explanations (not accuracy) in CNNs (Strout et al., 2019).

**Other ways to constrain DNNs**   While we focus on the use of explanations to constrain the relationships learned by neural networks, other approaches for constraining neural networks have also been proposed. A computationally intensive alternative is to augment the dataset in order to prevent the model from learning undesirable relationships, through domain knowledge (Bolukbasi et al., 2016), projecting out superficial statistics (Wang et al., 2019) or dramatically altering training images (Geirhos et al., 2018). However, these processes are often not feasible, either due to their computational cost or the difficulty of constructing such an augmented data set. Adversarial training has also been explored (Zhang & Zhu, 2019). These techniques are generally limited, as they are often tied to particular datasets, and do not provide a clear link between learning about a model's learned relationships through explanations, and subsequently correcting them.

## 3 METHODS

We now introduce CDEP, which penalizes the explanations of a neural network in order to align with prior knowledge about why a model should make a prediction. To do so, for each data point

it penalizes the CD scores of features, or groups of features, which a user does not want the model to learn to be important. While we focus on CD scores, which allow the penalization of interactions between features in addition to features themselves, this approach readily generalizes to other interpretation techniques, so long as they are differentiable.

## 3.1 AUGMENTING THE LOSS FUNCTION

Given a particular classification task, we want to teach a model to not only produce the correct prediction, but also to arrive at the prediction for the correct reasons. That is, we want the model to be right for the right reasons, where the right reasons are provided by the user and are dataset-dependent.

To accomplish this, CDEP modifies the objective function used to train a neural network, as displayed in Eq 1. In addition to the standard prediction loss $\mathcal{L}$, which teaches the model to produce the correct predictions, CDEP adds an explanation error $\mathcal{L}_{\text{expl}}$, which teaches the model to produce the correct explanations for its predictions. In place of the prediction and labels $f_\theta(X), y$, used in the prediction error $\mathcal{L}$, the explanation error $\mathcal{L}_{\text{expl}}$ uses the explanations produced by an interpretation method $\text{expl}_\theta(X)$, along with targets provided by the user $\text{expl}_X$. As is common with penalization, the two losses are weighted by a hyperparameter $\lambda \in \mathbb{R}$:

$$\hat{\theta} = \underset{\theta}{\text{argmin}} \ \underbrace{\mathcal{L}\left(f_\theta(X), y\right)}_{\text{Prediction error}} + \lambda \underbrace{\mathcal{L}_{\text{expl}}\left(\text{expl}_\theta(X), \text{expl}_X\right)}_{\text{Explanation error}} \tag{1}$$

The precise meanings of $\text{expl}_X$ depend on the context. For example, in the skin cancer image classification task described in Section 4, many of the benign skin images contain band-aids, but none of the malignant images. To force the model to ignore the band-aids in making their prediction, in each image $\text{expl}_\theta(X)$ denotes the importance score of the band-aid and $\text{expl}_X$ would be zero. These and more examples are further explored in Section 4.

## 3.2 CONTEXTUAL DECOMPOSITION (CD)

In this work, we use the CD score as the explanation function. In contrast to other interpretation methods, which focus on feature importances, CD also captures interactions between features. CD was originally designed for LSTMs (Murdoch et al., 2018) and subsequently extended to convolutional neural networks and arbitrary DNNs (Singh et al., 2018). For a given DNN $f(x)$, one can represent its output as a SoftMax operation applied to logits $g(x)$. These logits, in turn, are the composition of $L$ layers $g_i$, such as convolutional operations or ReLU non-linearities.

$$f(x) = \text{SoftMax}(g(x)) = \text{SoftMax}(g_L(g_{L-1}(...(g_2(g_1(x)))))) \tag{2}$$

Given a group of features $\{x_j\}_{j \in S}$, the CD algorithm, $g^{CD}(x)$, decomposes the logits $g(x)$ into a sum of two terms, $\beta(x)$ and $\gamma(x)$. $\beta(x)$ is the importance score of the feature group $\{x_j\}_{j \in S}$, and $\gamma(x)$ captures contributions to $g(x)$ not included in $\beta(x)$. The decomposition is computed by iteratively applying decompositions $g_i^{CD}(x)$ for each of the layers $g_i(x)$.

$$g^{CD}(x) = g_L^{CD}(g_{L-1}^{CD}(...(g_2^{CD}(g_1^{CD}(x))))) = (\beta(x), \gamma(x)) \tag{3}$$

$$\beta(x) + \gamma(x) = g(x) \tag{4}$$

## 3.3 CDEP OBJECTIVE FUNCTION

We now substitute the above CD scores into the generic equation in Eq 1 to arrive at the method used in this paper. While we use CD for the explanation method $\text{expl}_\theta(X)$, other explanation methods could be readily substituted at this stage. In order to convert CD scores to probabilities, we apply a SoftMax operation to $g^{CD}(x)$, allowing for easier comparison with the user-provided labels $\text{expl}_X$. We collect from the user, for each input $x_i$, a collection of feature groups $x_{i,S}$, $x_i \in \mathbb{R}^d, S \subseteq \{1, ..., d\}$, along with explanation target values $\text{expl}_{x_{i,S}}$, and use the $\|\cdot\|_1$ loss for $\mathcal{L}_{\text{expl}}$.

$$\hat{\theta} = \underset{\theta}{\arg\min} \underbrace{\sum_i \sum_c - y_{i,c} \log f_\theta(x_i)_c}_{\text{Classification error}} + \lambda \underbrace{\sum_i \sum_S ||\beta(x_{i,S}) - \text{expl}_{x_{i,S}}||_1}_{\text{Explanation error}} \quad (5)$$

In the above, $i$ indexes each individual example in the dataset, $S$ indexes a subset of the features for which we penalize their explanations, and $c$ sums over each class. Updating the model parameters in accordance with this formulation ensures that the model not only predicts the right output but also does so for the right (aligned with prior knowledge) reasons.

### 3.4 COMPUTATIONAL CONSIDERATIONS

A similar idea to Eq 1 has been proposed in previous/concurrent work, where the choice of explanation method uses a gradient-based attribution method (Ross et al., 2017; Erion et al., 2019). However, using such methods leads to three main complications which are solved by our approach. The first complication is the optimization process. When optimizing over attributions from a gradient-based attribution method via gradient descent, the optimizer requires the gradient of the gradient, thus requiring that all network components be twice differentiable. This process is computationally expensive and indeed optimizing it exactly involves optimizing over a differential equation. In contrast, CD attributions are calculated along with the forward pass of the network, and as a result can be optimized plainly with back-propagation using the standard single forward-pass and backward-pass per batch.

A second complication solved by the use of CD in Eq 5 is the ability to quickly finetune a pre-trained network. In many applications, particularly in transfer learning, it is common to finetune only the last few layers of a pre-trained neural network. Using CD, one can freeze early layers of the network and then finetune the last few layers of the network quickly as the activations and gradients of the frozen layers are not necessary.

Third, penalizing gradient-based methods incurs a very large memory usage. Using gradient-based methods, training requires the storage of activations and gradients for all layers of the network as well as the gradient of input (which can be omitted in normal training). Even for the simplest version, based on saliency, this more than doubles the required memory for a given batch and network size. More advanced methods proved to be completely infeasible to apply to a real-life dataset used, since the memory requirements were too high. By contrast, penalizing CD only requires a small constant amount of memory more than standard training.

## 4 RESULTS

The results here demonstrate the efficacy of CDEP on a variety of datasets using diverse explanation types. Sec 4.1 shows results on ignoring spurious patches in the ISIC skin cancer dataset (Codella et al., 2019), Sec 4.2 details experiments on converting a DNN's preference for color to a preference for shape on a variant of the MNIST dataset (LeCun, 1998), and Sec 4.3 shows experiments on text data from the Stanford Sentiment Treebank (SST) (Socher et al., 2013).[1]

### 4.1 IGNORING SPURIOUS SIGNALS IN SKIN CANCER DIAGNOSIS

In recent years, deep learning has achieved impressive results in diagnosing skin cancer, with predictive accuracy sometimes comparable to human doctors (Esteva et al., 2017). However, the datasets used to train these models often include spurious features which make it possible to attain high test accuracy without learning the underlying phenomena (Winkler et al., 2019). In particular, a popular dataset from ISIC (International Skin Imaging Collaboration) has colorful patches present in approximately 50% of the non-cancerous images but not in the cancerous images (Codella et al., 2019). An unpenalized DNN learns to look for these patches as an indicator for predicting that an image is benign. We use CDEP to remedy this problem by penalizing the DNN placing importance on the patches during training.

---

[1] All models were trained in PyTorch.

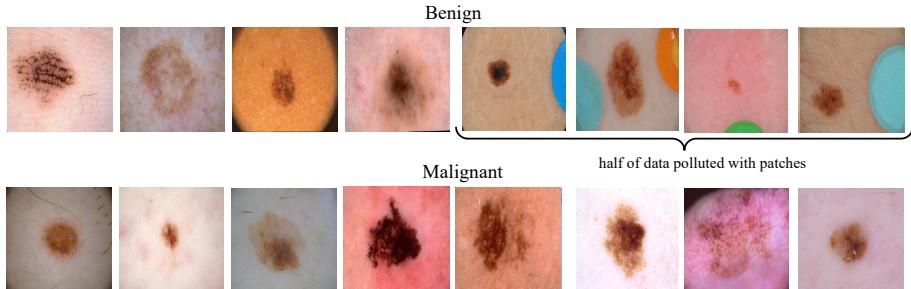

Figure 2: Example images from the ISIC dataset. Half of the benign lesion images include a patch in the image.

The task in this section is to classify whether an image of a skin lesion contains (1) benign melanoma or (2) malignant melanoma. The ISIC dataset consists of 21,654 images (19,372 benign), each diagnosed by histopathology or a consensus of experts. For classification, we use a VGG16 architecture (Simonyan & Zisserman, 2014) pre-trained on the ImageNet Classification task [2] and freeze the weights of early layers so that only the fully connected layers are trained. In order to use CDEP, the spurious patches are identified via a s imple image segmentation algorithm using a color threshold (see Sec S4).

Table 1 shows results comparing the performance of a model trained with and without CDEP. We report results on two variants of the test set. The first, which we refer to as "no patches" only contains images of the test set that do not include patches. The second also includes images with those patches. Training with CDEP improves the AUC and F1-score for both test sets.

Table 1: Results from training a DNN on ISIC to recognize skin cancer (averaged over three runs). Results shown for the entire test set and for only the images the test set that do not include patches ("no patches"). The network trained with CDEP generalizes better, getting higher AUC and F1 on both. Std below 0.006 for all AUC and below 0.012 for all F1.

|  | AUC (no patches) | F1 (no patches) | AUC (all) | F1 (all) |
| --- | --- | --- | --- | --- |
| Vanilla (excluded data) | 0.86 | 0.59 | 0.92 | 0.59 |
| Vanilla | 0.85 | 0.56 | 0.92 | 0.56 |
| With RRR | 0.66 | 0.39 | 0.82 | 0.39 |
| With CDEP | **0.88** | **0.61** | **0.93** | **0.61** |

In the first row of Table 1, the model is trained using only the data without the spurious patches, and the second row shows the model trained on the full dataset. The network trained using CDEP achieves the best AUC, surpassing both unpenalized versions. Applying our method increases the ROC AUC as well as the best F1 score. We also compared our method against the method introduced in 2017 by Ross et al. (RRR). For this, we restricted the batch size to 16 (and consequently use a learning rate of $10^{-5}$) due to memory constraints. Using RRR did not improve on the base AUC, implying that penalizing gradients is not helpful in penalizing higher-order features.[3]

**Visualizing explanations** Fig. 3 visualize GradCAM heatmaps (Ozbulak, 2019; Selvaraju et al., 2017) for an unpenalized DNN and a DNN trained with CDEP to ignore spurious patches. As expected, after penalizing with CDEP, the DNN attributes less importance to the spurious patches, regardless of their position in the image. More examples, also for cancerous images, are shown in Sec S5.

---

[2]Pre-trained model retrieved from torchvision.

[3]We were not able to compare against the method recently proposed in Erion et al. (2019) due to the prohibitively slow training and large memory requirements.

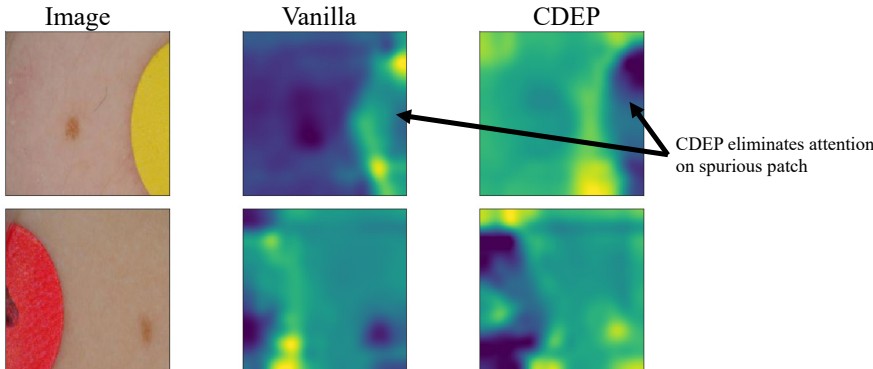

Figure 3: Visualizing heatmaps for correctly predicted exampes from the ISIC skincancer test set. Lighter regions in the heatmap are attributed more importance. The DNN trained with CDEP correctly captures that the patch is not relevant for classification.

## 4.2 COMBATING INDUCTIVE BIAS ON VARIANTS OF THE MNIST DATASET

In this section, we investigate whether we can alter which features a DNN uses to perform digit classification, using variants of the MNIST dataset (LeCun, 1998) and a standard CNN architecture for this dataset retrieved from PyTorch [4].

### 4.2.1 COLORMNIST

Similar to a previous study (Li & Vasconcelos, 2019), we transform the MNIST dataset to include three color channels and assign each class a distinct color, as shown in Fig. 4. An unpenalized DNN trained on this biased data will completely misclassify a test set with inverted colors, dropping to 0% accuracy (see Section 4.2.1), suggesting that it learns to classify using the colors of the digits rather than their shape.

Here, we want to see if we can alter the DNN to focus on the shape of the digits rather than their color. Interestingly, this can be enforced by minimizing the contribution of pixels in isolation while maximizing the importance of groups of pixels (which can represent shapes). To do this, we add penalize the CD contribution of sampled single pixel values, following Eq 5. By minimizing the contribution of single pixels we effectively encourage the network to focus more on groups of pixels, which can represent shape.

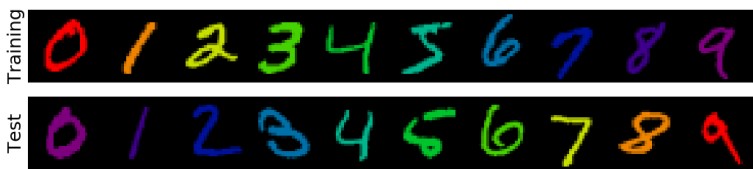

Figure 4: ColorMNIST: the test set shapes remain the same as the training set, but the colors are inverted. A vanilla network trained on this training set will get 0% accuracy on the test set.

Section 4.2.1 shows that CDEP can partially change the network's focus on solely color to also focus on digit shape. We compare CDEP to two previously introduced explanation penalization techniques: penalization of the squared gradients (RRR) (Ross et al., 2017) and Expected Gradients (EG) (Erion et al., 2019). For EG we penalize variance between attributions of the RGB channels

---

[4]Model and training code from https://github.com/pytorch/examples/blob/master/mnist/main.py.

(as recommended by the authors of EG in personal correspondence). None of the baselines are able to improve the test accuracy of the model on this task above the random baseline, while CDEP is able to significantly improve this accuracy to 31.0%. We show the increase of predictive accuracy with increasing penalization in Fig. S5.

Table 2: Results on ColorMNIST (test accuracy). All values averaged over thirty runs. CDEP is the only method that captures and removes color bias.

|  | Unpenalized | Baseline | CDEP | RRR | Expected Gradients |
|---|---|---|---|---|---|
| Test Accuracy | $0.2 \pm 0.2$ | 9.8 | $\mathbf{31.0 \pm 2.3}$ | $0.2 \pm 0.1$ | $10.0 \pm 0.1$ |

### 4.2.2 DECOYMNIST

For further comparison with previous work, we evaluate CDEP on an existing task: DecoyMNIST (Erion et al., 2019). DecoyMNIST adds a class-indicative gray patch to a random corner of the image. This task is relatively simple, as the spurious features are not entangled with any other feature and are always at the same location (the corners). Table 3 shows that all methods perform roughly equally, recovering the base accuracy. Results are reported using the best penalization parameter $\lambda$, chosen via cross-validation on the test accuracy. We provide details on the computation time, and memory usage in Table S1, showing that CDEP is similar to existing approaches. However, when freezing early layers of a network and finetuning, CDEP very quickly becomes more efficient than other methods.

Table 3: Results on Grayscale Decoy set.

|  | Unpenalized | CDEP | RRR | Expected Gradients |
|---|---|---|---|---|
| Test accuracy | $60.1 \pm 5.1$ | $97.2 \pm 0.8$ | $99.0 \pm 1.0$ | $97.8 \pm 0.2$ |

### 4.3 FIXING BIAS IN TEXT DATA

To demonstrate CDEP's effectiveness on text, we use the Stanford Sentiment Treebank (SST) dataset (Socher et al., 2013), an NLP benchmark dataset consisting of movie reviews with a binary sentiment (positive/negative). We inject spurious signals into the training set and train a standard LSTM [5] to classify sentiment from the review.

### Positive

pacino is the best **she**'s been in years and keener is marvelous

**she** showcases davies as a young woman of great charm , generosity and diplomacy

shows **she** 's back in form , with an astoundingly rich film .

proves once again that **she**'s the best brush in the business

### Negative

green ruins every single scene **he**'s in, and the film, while it 's not completely wreaked, is seriously compromised by that

i'm sorry to say that this should seal the deal - arnold is not, nor will **he** be, back .

this is sandler running on empty , repeating what **he** 's already done way too often .

so howard appears to have had free rein to be as pretentious as **he** wanted

Figure 5: Example sentences from the SST dataset with artificially induced bias on gender.

---

[5]Model and training code from https://github.com/clairett/pytorch-sentiment-classification.

We create three variants of the SST dataset, each with different spurious signals which we aim to ignore (examples in Sec S1). In the first variant, we add indicator words for each class (positive: 'text', negative: 'video') at a random location in each sentence. An unpenalized DNN will focus only on those words, dropping to nearly random performance on the unbiased test set. In the second variant, we use two semantically similar words ('the', 'a') to indicate the class by using one word only in the positive and one only in the negative class. In the third case, we use 'he' and 'she' to indicate class (example in Fig 5). Since these gendered words are only present in a small proportion of the training dataset ($\sim 2\%$), for this variant, we report accuracy only on the sentences in the test set that do include the pronouns (performance on the test dataset not including the pronouns remains unchanged). Table 4 shows the test accuracy for all datasets with and without CDEP. In all scenarios, CDEP is successfully able to improve the test accuracy by ignoring the injected spurious signals.

Table 4: Results on SST. CDEP substantially improves predictive accuracy on the unbiased test set after training on biased data.

|  | Unpenalized | CDEP |
| --- | --- | --- |
| Random words | $56.6 \pm 5.8$ | **$75.4 \pm 0.9$** |
| Biased (articles) | $57.8 \pm 0.8$ | **$68.2 \pm 0.8$** |
| Biased (gender) | $64.2 \pm 3.1$ | **$78.0 \pm 3.0$** |

## 5 CONCLUSION

In this work we introduce a novel method to penalize neural networks to align with prior knowledge. Compared to previous work, CDEP is the first of its kind that can penalize complex features and feature interactions. Furthermore, CDEP is more computationally efficient than previous work and does not rely on backpropagation, enabling its use with more complex neural networks. We show that CDEP can be used to remove bias and improve predictive accuracy on a variety of toy and real data. The experiments here demonstrate a variety of ways to use CDEP to improve models both on real and toy datasets. CDEP is quite versatile and can be used in many more areas to incorporate the structure of domain knowledge (e.g. biology or physics). Of course, the effectiveness of CDEP depends upon the quality of the prior knowledge used to determine the explanation targets. Future work includes extending CDEP to more complex penalties, incorporating more fine-grained explanations and interactions. We hope the work here will help push the field towards a more rigorous way to use interpretability methods, a point which will become increasingly important as interpretable machine learning develops as a field (Doshi-Velez & Kim, 2017; Murdoch et al., 2019).

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

# Supplement

## S1    ADDITIONAL DETAILS ABOUT SST TASK

Section 4.3 shows the results for CDEP on biased variants of the SST dataset. Here we show examples of the biased sentences (for task 2 and 3 we only show sentences where the bias was present) in Figs. S1 to S3. For the first task, we insert two randomly chosen words in 100% of the sentences in the positive and negative class respectively. We choose two words ("text" for the positive class and "video" for the negative class) that were not otherwise present in the data set but had a representation in Word2Vec.

**Positive**

part of the charm of satin rouge is that it avoids the obvious with **text** humour and lightness .

**text** a screenplay more ingeniously constructed than 'memento'

good fun **text**, good action, good acting, good dialogue, good pace, good cinematography .

dramas like **text** this make it human .

**Negative**

... begins with promise, but runs aground after being **video** snared in its own tangled plot .

the **video** movie is well done, but slow .

this orange has some juice , but it 's **video** far from fresh-squeezed .

as it is, **video** it 's too long and unfocused .

Figure S1: Example sentences from the variant 1 of the biased SST dataset with decoy variables in each sentence.

For the second task, we choose to replace two common words ("the" and "a") in sentences where they appear (27% of the dataset). We replace the words such that one word only appears in the positive class and the other world only in the negative class. By choosing words that are semantically almost replaceable, we ensured that the normal sentence structure would not be broken such as with the first task.

**Positive**

comes off as **a** touching , transcendent love story .

is most remarkable not because of its epic scope , but because of **a** startling intimacy

couldn't be better as **a** cruel but weirdly likable wasp matron

uses humor and **a** heartfelt conviction to tell that story about discovering your destination in life

**Negative**

to creep **the** living hell out of you

holds its goodwill close , but is relatively slow to come to **the** point

it 's not **the** great monster movie .

consider **the** dvd rental instead

Figure S2: Example sentences from the variant 2 of the SST dataset with artificially induced bias on articles ("the", "a"). Bias was only induced on the sentences where those articles were used (27% of the dataset).

For the third task we repeat the same procedure with two words ("he" and "she") that appeared in only 2% of the dataset. This helps evaluate whether CDEP works even if the spurious signal appears only in a small section of the data set.

**Positive**

pacino is the best **she**'s been in years and keener is marvelous

**she** showcases davies as a young woman of great charm , generosity and diplomacy

shows **she** 's back in form , with an astoundingly rich film .

proves once again that **she**'s the best brush in the business

**Negative**

green ruins every single scene **he**'s in, and the film, while it 's not completely wreaked, is seriously compromised by that

i'm sorry to say that this should seal the deal - arnold is not, nor will **he** be, back .

this is sandler running on empty , repeating what **he** 's already done way too often .

so howard appears to have had free rein to be as pretentious as **he** wanted

Figure S3: Example sentences from the variant 3 of the SST dataset with artificially induced bias on articles ("he", "she"). Bias was only induced on the sentences where those articles were used (2% of the dataset).

## S2 NETWORK ARCHITECTURES AND TRAINING

### S2.1 NETWORK ARCHITECTURES

For the ISIC skin cancer task we used a pretrained VGG16 network retrieved from the PyTorch model zoo. We use SGD as the optimizer with a learning rate of 0.01 and momentum of 0.9. Preliminary experiments with Adam as the optimizer yielded poorer predictive performance.

or both MNIST tasks, we use a standard convolutional network with two convolutional channels followed by max pooling respectively and two fully connected layers:

Conv(20,5,5) - MaxPool() - Conv(50,5,5) - MaxPool - FC(256) - FC(10). The models were trained with Adam, using a weight decay of 0.001.

Penalizing explanations adds an additional hyperparameter, $\lambda$ to the training. $\lambda$ can either be set in proportion to the normal training loss or at a fixed rate. In this paper we did the latter. We expect that exploring the former could lead to a more stable training process. For all tasks $\lambda$ was tested across a wide range between $[10^{-1}, 10^4]$.

The LSTM for the SST experiments consisted of two LSTM layers with 128 hidden units followed by a fully connected layer.

### S2.2 COLORMNIST

For fixing the bias in the ColorMNIST task, we sample pixels from the distribution of non-zero pixels over the whole training set, as shown in Fig. S4

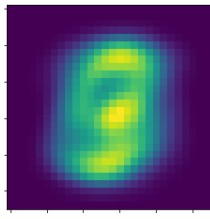

Figure S4: Sampling distribution for ColorMNIST

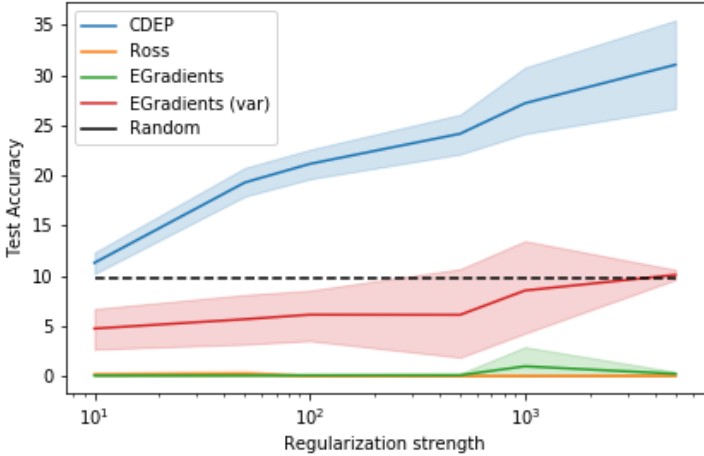

Figure S5: Results on ColorMNIST (Test Accuracy). All averaged over thirty runs. CDEP is the only method that captures and removes color bias.

For Expected Gradients we show results when sampling pixels as well as when penalizing the variance between attributions for the RGB channels (as recommended by the authors of EG) in Fig. S5. Neither of them go above random accuracy, only achieving random accuracy when they are regularized to a constant prediction.

## S3 RUNTIME AND MEMORY REQUIREMENTS OF DIFFERENT ALGORITHMS

This section provides further details on runtime and memory requirements reported in Table S1. We compared the runtime and memory requirements of the available regularization schemes when implemented in Pytorch.

Memory usage and runtime were tested on the DecoyMNIST task with a batch size of 64. It is expected that the exact ratios will change depending on the complexity of the used network and batch size (since constant memory usage becomes disproportionally smaller with increasing batch size).

The memory usage was read by recording the memory allocated by PyTorch. Since Expected Gradients and RRR require two forward and backward passes, we only record the maximum memory usage. We ran experiments on a single Titan X.

Table S1: Memory usage and run time were recorded for the DecoyMNIST task.

|  | Unpenalized | CDEP | RRR | Expected Gradients |
|---|---|---|---|---|
| Run time/epoch (seconds) | 4.7 | 17.1 | 11.2 | 17.8 |
| Maximum GPU RAM usage (GB) | 0.027 | 0.068 | 0.046 | 0.046 |

## S4 IMAGE SEGMENTATION FOR ISIC SKIN CANCER

To obtain the binary maps of the patches for the skin cancer task, we first segment the images using SLIC, a common image-segmentation algorithm (Achanta et al., 2012). Since the patches look quite distinct from the rest of the image, the patches are usually their own segment.

Subsequently we take the mean RGB and HSV values for all segments and filtered for segments which the mean was substantially different from the typical caucasian skin tone. Since different images were different from the typical skin color in different attributes, we filtered for those images

recursively. As an example, in the image shown in Fig. S6, the patch has a much higher saturation than the rest of the image. For each image we exported a map as seen in Fig. S6.

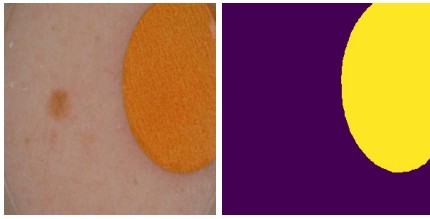

Figure S6: Sample segmentation for the ISIC task.

## S5  ADDITIONAL HEATMAP EXAMPLES FOR ISIC

We show additional examples from the test set of the skin cancer task in Figs. S7 and S8. We see that the importance maps for the unregularized and regularized network are very similar for cancerous images and non-cancerous images without patch. The patches are ignored by the network regularized with CDEP.

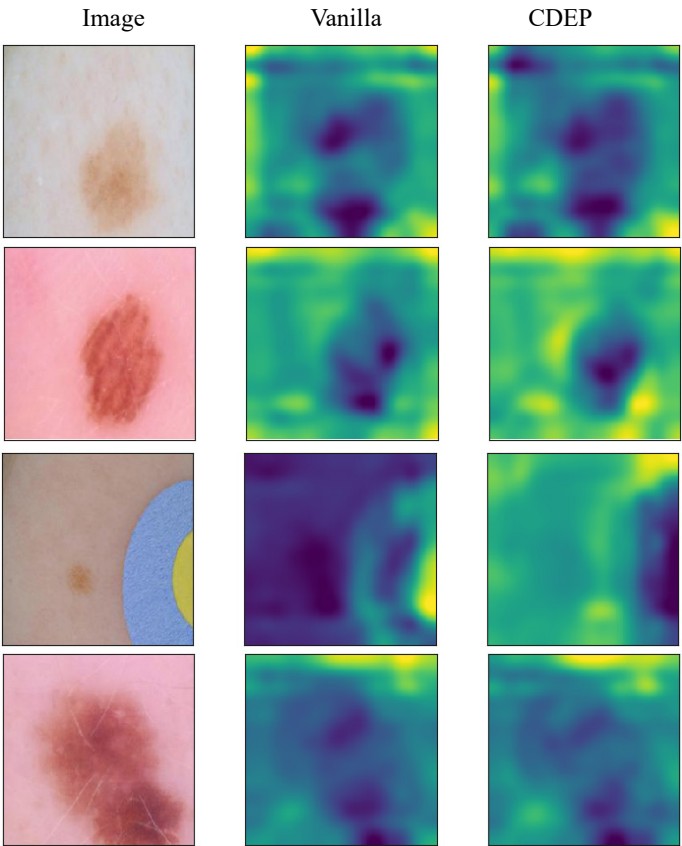

Figure S7: Heatmaps for benign samples from ISIC

Image          Vanilla          CDEP

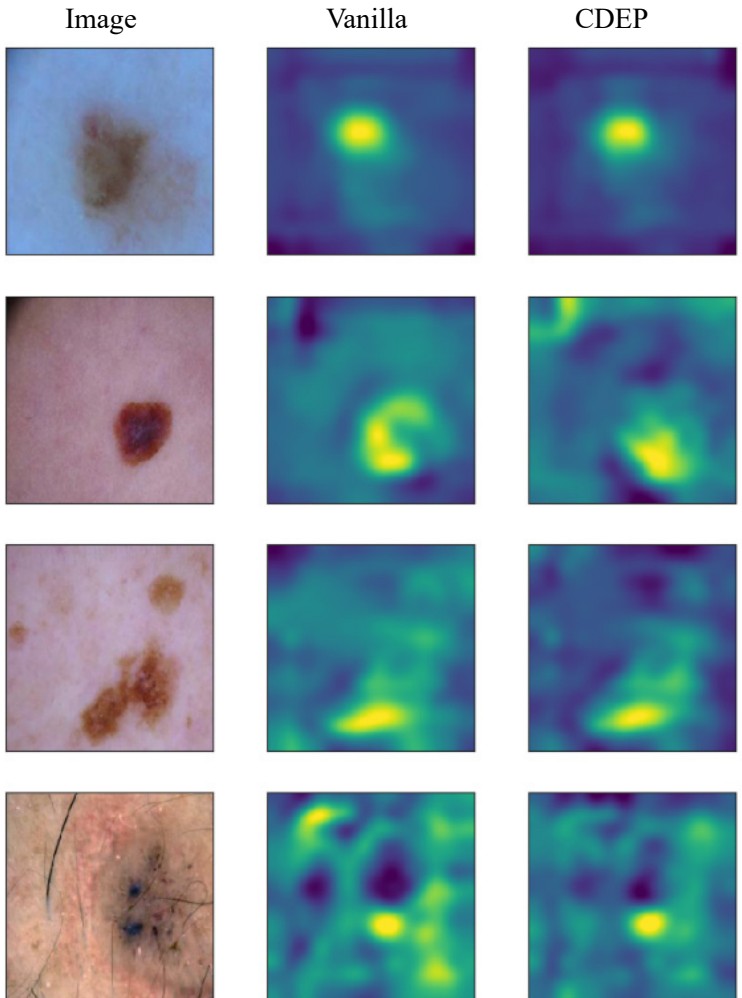

Figure S8: Heatmaps for cancerous samples from ISIC

A different spurious correlation that we noticed was that proportionally more images showing skin cancer will have a ruler next to the lesion. This is the case because doctors often want to show a reference for size if they diagnosed that the lesion is cancerous. Even though the spurious correlation is less pronounced (in a very rough cursory count, 13% of the cancerous and 5% of the benign images contain some sort of measure), the networks learnt to recognize and exploit this spurious correlation. This further highlights the need for CDEP, especially in medical settings.

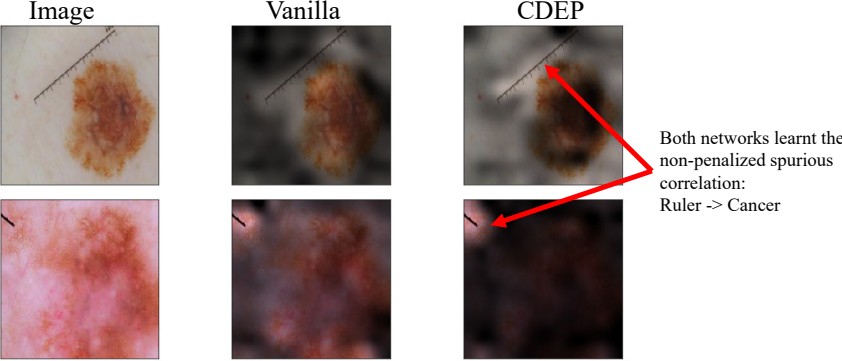

Figure S9: Both networks learnt that proportionally more images with malignant lesions feature a ruler next to the lesion. To make comparison easier, we visualize the heatmap by multiplying it with the image. Visible regions are important for classification.

