# OpenReview forum: "Interpretations are useful: penalizing explanations to align neural networks with prior knowledge"
_ICLR.cc/2020/Conference — Reject_

### Official Review · AnonReviewer3 · 2019-10-22
**Official Blind Review #3**

**Rating:** 6

**Review:**

This paper presents a method intended to allow practitioners to *use* explanations provided by various methods. Concretely, the authors propose contextual decomposition explanation penalization (CDEP), which aims to use explanation methods to allow users to dissuade the model from learning unwanted correlations.

The proposed method is somewhat similar to prior work by Ross et al., in that the idea is to include an explicit term in the objective that encourages the model to align with prior knowledge. In particular, the authors assume supervision --- effectively labeled features, from what I gather --- provided by users and define an objective that penalizes divergence from this. The object that is penalized is $\Beta(x_i, s)$, which is the importance score for feature s in instance $i$; for this they use a decontextualized representation of the feature (this is the contextual decomposition aspect). Although the authors highlight that any differentiable scoring function could be used, I think the use of this decontextualized variant as is done here is nice because it avoids issues with feature interactions in the hidden space that might result in misleading 'attribution' w.r.t. the original inputs.

The main advantage of this effort compared to work that directly penalizes the gradients (as in Ross et al.) is that the method does not rely on second gradients (gradients of gradients), which is computationally problematic. Overall, this is a nice contribution that offers a new mechanism for exploiting human provided annotations. I do have some specific comments below.

- I am not sure I agree with the premise as stated here. Namely, the authors write "For an explanation of a deep learning model to be effective, it must provide both insight into a model and suggest a corresponding action in order to achieve some objective" -- I would argue that an explanation may be useful in and of itself by highlighting how a model came to a prediction. I am not convinced that it need necessarily lead to, e.g., improving model performance. I think the authors are perhaps arguing that explanations might be used to interactively improve the underlying model, which is an interesting and sensible direction.

- This work, which aims to harness user supervision on explanations to improve model performance, seems closely related to work on "annotator rationales" (Zaidan 2007 being the first work on this), but no mention is made of this. "Do Human Rationales Improve Machine Explanations?" by Strout et al. (2019) also seems relevant as a more recent instance in this line of work. I do not think such approaches are necessarily directly comparable, but some discussion of how this effort is situatied with respect to this line of work would be appreciated.

- The experiment with MNIST colors was neat.

- The authors compare their approach to Ross and colleagues in Table 1 but see quite poor results for the latter approach. Is this a result of the smaller batch size / learning rate adjustment? It seems that some tuning of this approach is warranted.

- Figure 3 is nice but not terribly surprising: The image shows that the objective indeed works as expected; but if this were not the case, then it would suggest basically a failure of optimization (i.e., the objective dictates that the image should look like this *by construction*). Still, it's a good sanity check.




**Experience Assessment:**

I have published one or two papers in this area.

**Review Assessment: Checking Correctness Of Derivations And Theory:**

I assessed the sensibility of the derivations and theory.

**Review Assessment: Checking Correctness Of Experiments:**

I assessed the sensibility of the experiments.

**Review Assessment: Thoroughness In Paper Reading:**

I read the paper at least twice and used my best judgement in assessing the paper.

---

> ### Author Response · Authors · 2019-11-07
> **Author's response**
>
> We would like to thank the reviewer for their thoughtful review. We address your concerns below.
>
> “The main advantage of this effort compared to work that directly penalizes the gradients (as in Ross et al.) is that the method does not rely on second gradients (gradients of gradients), which is computationally problematic”
>
> While our approach has computational benefits, we would also note that empirically CDEP produces significantly better results. On color MNIST Ross et al. provides no benefit (accuracy the same as a random baseline - 10%), while CDEP achieves 31%. Similarly, for the skin cancer dataset, Ross et al. actually hurt accuracy. We attribute this to CDEP allowing the penalization of features, including interactions of features, rather than just feature-level gradients.
>
> “I am not sure I agree with the premise as stated here. Namely, the authors write "For an explanation of a deep learning model to be effective, it must provide both insight into a model and suggest a corresponding action in order to achieve some objective" -- I would argue that an explanation may be useful in and of itself by highlighting how a model came to a prediction. I am not convinced that it need necessarily lead to, e.g., improving model performance. I think the authors are perhaps arguing that explanations might be used to interactively improve the underlying model, which is an interesting and sensible direction.”
>
> We agree that our abstract is strongly worded - this is by design. We feel that explainable deep learning research is currently overwhelmed with different explanation algorithms, yet has very few (arguably no) success stories of researchers actually using these algorithms to accomplish something of interest to the broader community.
>
> Explainable DL techniques can certainly be used to “highlight how a model came to a prediction”, but we feel that this is only an intermediate objective, not an end in itself. Ultimately, users want to do things like improve model performance, build trust in a model, identify flaws, or verify that model is being fair with respect to attributes like race, gender, etc.
>
> As a community, we do not currently know how to use our explanation algorithms to accomplish these things, or whether our explanation algorithms are well suited to do so. In fact, we suspect that many published explanation algorithms would fail when evaluated on real end tasks - as we saw with gradients and integrated gradients in this paper.
>
> Figuring out how to use explainable DL techniques for anything real is essentially being neglected by current researchers, so we framed our paper to try to shed light on this, and nudge things in the right direction.
>
> If you still disagree with our premise, we’d be happy to tamp things down, and adjust our abstract to motivate things through the vein of “explanations could be useful to improve predictions”.
>
> “This work, which aims to harness user supervision on explanations to improve model performance, seems closely related to work on "annotator rationales" (Zaidan 2007 being the first work on this), but no mention is made of this. "Do Human Rationales Improve Machine Explanations?" by Strout et al. (2019) also seems relevant as a more recent instance in this line of work“
>
> Thanks for bringing Zaidan 2007 and Strout 2019 to our attention, they are indeed useful prior work in this field. We will include both references in an updated version.
>
> “The authors compare their approach to Ross and colleagues in Table 1 but see quite poor results for the latter approach. Is this a result of the smaller batch size / learning rate adjustment? It seems that some tuning of this approach is warranted.”
>
> As you noted, the approach by Ross penalizes gradients of gradients, preventing learning of those weights. This works quite nicely for tasks where the feature to be ignored is always in the same location as we see in the results on DecoyMNIST.  In contrast, for the ISIC dataset, the patches are distributed roughly uniformly over the image.  By penalizing gradients for the patches, the gradient updates are ‘dampened’ over the entire input for a large part of the training data (patches are present in 45% of samples) and learning is prevented. This issue may be further amplified by the low learning rate and batch size necessary for this approach and dataset.
>
> We can assure you we tried our best to tune Ross’ approach in order to achieve a fair baseline (despite the fact that their approach is roughly 80 times slower than CDEP, making extensive tuning difficult).

---

> > ### Author Response · Authors · 2019-11-07
> > **Author's response continued**
> >
> > “Figure 3 is nice but not terribly surprising”
> >
> > We agree - we included Figure 3 not as a shocking finding, but to visually explain what our method does (in addition to the text/equation descriptions elsewhere), as well as provide a sanity check. It should also be noted that we obtain the explanations with a different method (GradCAM) than the one used for the optimization (CD). This gave us some indication that we were not overfitting to a particular explanation algorithm.

---

### Official Review · AnonReviewer2 · 2019-10-23
**Official Blind Review #2**

**Rating:** 3

**Review:**

The paper presents a way of using generated explanations of model predictions to help prevent a model from learning "unwanted" relationships between features and class labels. This idea was implemented with a particular explanation generation method from prior work, called contextual decomposition (CD). For a given feature, the corresponding CD can be used to measure its importance. The proposed learning objective in this work optimizes not only the cross entropy loss, but also the difference between the CD score of a given feature and its explanation target value. Experiments show that this new learning algorithm can largely improve the classification performance.

I like the high-level idea of this work and agree that there is not much work on using prediction explanations to help improve model performance. However, there are two major concerns of the model and experiment design.

First, it seems like the proposed method requires whoever use it already know what the problem is. For example,

- in section 3.3, the model inputs include a collection of features and the corresponding explanation target values.
- in section 4.1, it is already known that some colorful patches only appear in some non-cancerous images but not in cancerous images.
- it is even more obvious in section 4.2 and 4.3, because in both experiments, the training and test examples were altered on purpose to create some mismatch.

My question is that if we already know the bias or the mismatch, why not directly use this information in the regularization to penalize some features? Is it necessary to resort to some explanation generation methods?

My second concern is more like a personal opinion. In the experiment of section 4.2, if the colors are good indicators of these digits in the training set, I don't it is wrong for a model to capture these important features. However, the way of altering examples in the same class with different colors in training and test sets seems questionable, because now, the distributions of training and test images are different. On the other hand, if we already know color is the issue, why not simply convert the images into black-and-white? A similar argument can also be applied to the experiment in section 4.3

Overall, I like the idea of using explanations to help build a better classifier. However, I am concerned about the value of this work.


**Experience Assessment:**

I have published in this field for several years.

**Review Assessment: Checking Correctness Of Derivations And Theory:**

I carefully checked the derivations and theory.

**Review Assessment: Checking Correctness Of Experiments:**

I carefully checked the experiments.

**Review Assessment: Thoroughness In Paper Reading:**

I read the paper thoroughly.

---

> ### Author Response · Authors · 2019-11-07
> **Author's response**
>
> We would like to thank the reviewer for their time and thoughtful comments. We address their concerns below.
>
> “I like the high-level idea of this work and agree that there is not much work on using prediction explanations to help improve model performance. However, there are two major concerns of the model and experiment design.
>
> First, it seems like the proposed method requires whoever use it already know what the problem is.”
>
> We agree - after a practitioner has found a flaw in their model or limitations in their training data (using any existing interpretation technique), our technique is designed to help rectify that flaw by altering the model.
>
> “My question is that if we already know the bias or the mismatch, why not directly use this information in the regularization to penalize some features? Is it necessary to resort to some explanation generation methods?”
>
> It is not clear to us how, exactly, we could “directly use this information in the regularization to penalize features” without resorting to explanations. Consider the skin cancer (ISIC) example shown in Figure 2. The image patches that we want the model to ignore occur at different places in different images, and the model used is a CNN. To us, it is unclear how to compute (let alone regularize) the contribution of a feature to a model/prediction without the use of explanations. Beyond the methods we compare against, we are not aware of any other way to do so.
>
> If there is relevant prior work that we are missing that describes such a technique, we would love to take a look.
>
> “My second concern is more like a personal opinion. In the experiment of section 4.2, if the colors are good indicators of these digits in the training set, I don't it is wrong for a model to capture these important features. However, the way of altering examples in the same class with different colors in training and test sets seems questionable, because now, the distributions of training and test images are different. On the other hand, if we already know color is the issue, why not simply convert the images into black-and-white? A similar argument can also be applied to the experiment in section 4.3”
>
> This is a great thought, which merits some discussion. At first blush, we agree that these are simple problems, which could be solved in simpler ways.
>
> However, we feel these simulations studies are actually very useful in developing and evaluating algorithms (including, but not limited to explanation algorithms). The reason for this is that they present a “bare minimum” for prospective methods to clear, and provide a clear metric of success. Put simply, if a prospective method cannot solve something so simple, it is unlikely to be of use on any “real”, i.e. not simulated, datasets.
>
> In the color MNIST example of section 4.2, for instance, we have a clearly defined, spurious correlation (color), which is easy to check if a method has successfully removed from a model. In this idealized setting, our method was able to partially remove the confounding, while other techniques fail completely (underperforming a random benchmark).
>
> Of course, passing the “bare minimum” is not sufficient to fully validate a method, which is why we included a very real and consequential example in section 4.1 on skin cancer detection. However, we do think that CDEP’s performance on simulations (both absolute and relative to baselines) provides additional, meaningful, evidence of its effectiveness.
>
> As an aside, we are far from the first ones to use simulation studies like this to validate our methods. Color MNIST was in CVPR last year [1], and was also discussed in a keynote at ICLR 2019 (this is what led us to use it). The Decoy-MNIST dataset was introduced in [2], a fairly successful (90+ citation in 2 years) paper.
>
> [1] https://arxiv.org/pdf/1904.07911.pdf
> [2] https://arxiv.org/pdf/1703.03717.pdf

---

### Official Review · AnonReviewer1 · 2019-10-25
**Official Blind Review #1**

**Rating:** 3

**Review:**

The authors propose to add a regularizer to the loss function when training a prediction model. In particular, the regularizer considers explanations during the model training; if the explanations are not consistent with some prior knowledge, then explanation errors will be introduced.

The motivation for the proposed research is interesting and has some merit. However, I am a bit worried that the proposed approach is somewhat ad hoc. I can imagine there are various explanations that can be generated for the same model. There can also be different prior knowledge available for a particular problems. Which prior knowledge and explanations to use seem to affect a lot about the learned model. But there is no principled approaches for making the selection.

In some sense, standard regularizers such as L1 or L2 are are intrinsic regularizers, while the proposed regularizer is extrinsic regularizer. I think the extrinsic regularizer certainly has some merit, but it is also hard to regulate.

For instance, consider the example in Figure 2 about the presence of patches. Isn't that a too specific knowledge about the dataset, which in turn makes the proposed approach not general? I have doubts on how useful a method is if it relies on such specific prior knowledge about the data.

**Experience Assessment:**

I have read many papers in this area.

**Review Assessment: Checking Correctness Of Derivations And Theory:**

I assessed the sensibility of the derivations and theory.

**Review Assessment: Checking Correctness Of Experiments:**

I assessed the sensibility of the experiments.

**Review Assessment: Thoroughness In Paper Reading:**

I read the paper at least twice and used my best judgement in assessing the paper.

---

> ### Author Response · Authors · 2019-11-07
> **Author's response**
>
> We would like to thank the reviewer for their thoughtful points. We have addressed their concerns below.
>
> “However, I am a bit worried that the proposed approach is somewhat ad hoc. I can imagine there are various explanations that can be generated for the same model. There can also be different prior knowledge available for a particular problems. Which prior knowledge and explanations to use seem to affect a lot about the learned model. But there is no principled approaches for making the selection.”
>
> This is an excellent point. In short, this is, in some sense of the word, an ad hoc method - but we don’t think that is a bad thing. CDEP’s ability to incorporate different forms of prior knowledge is a necessary feature to enable practitioners to use it in a wide variety of settings. While CDEP lacks a formal, mathematical derivation, it produces strong empirical results.
>
> “I can imagine there are various explanations that can be generated for the same model.”
>
> When it comes to the mechanical details of our algorithm, CDEP is certainly ad hoc. In particular, we have no proof that CDEP is mathematically optimal/unique, and it is possible that there could be some other version of CDEP, which could produce better results. Such a version may use a different explanation algorithm, or a different approach for penalizing the explanations.
>
> However, for the methodological choices we made, we are able to show meaningful empirical improvements across a number of different datasets. While a uniqueness proof would be nice, we feel that our empirical results are sufficient to demonstrate the effectiveness of our method.
>
> “There can also be different prior knowledge available for a particular problem”
>
> We should be clear - CDEP is not a plug and play tool that can be blindly applied without any knowledge of the underlying data. Rather, CDEP requires a practitioner to carefully examine their model, and dataset. Subsequently, CDEP enables them to use their best judgement in determining what patterns are likely to generalize, and should be used by the model.
>
> This type of “ad hoc” analysis is critical for real-world uses of machine learning, and CDEP provides a useful tool for doing so. In our skin cancer example, without properly analyzing the model and data, and using CDEP, a practitioner would construct a model that learns to predict whether a patient has a band-aid. Using that band-aid predictor to help diagnose skin cancer would be problematic, to say the least.
>
> “But there is no principled approaches for making the selection.”
>
> Practitioners can optimize their selections for predictive accuracy on an appropriate dataset.
>
>
> “For instance, consider the example in Figure 2 about the presence of patches. Isn't that a too specific knowledge about the dataset, which in turn makes the proposed approach not general? I have doubts on how useful a method is if it relies on such specific prior knowledge about the data.”
>
> As we discussed above, Figure 2 is one example of the type of prior knowledge that CDEP can use. However, the general theme of models learning spurious correlations is a fairly common problem that should not require much motivation.
>
> For other examples, within our paper, we’d point to our other results in 4.2 and 4.3, as well as prior work on penalizing explanations [1]. As noted by another reviewer, there is also a line of work on non-deep learning models in NLP surrounding annotator rationales [2] [3]. CDEP could also certainly be used in improving the fairness of a model (ensuring that a model does not discriminate based on sensitive attributes like gender, race, etc.). There have also been other failures in medical machine learning that could benefit from CDEP [4]. In the month since we completed this work, we’ve also come in touch with some biologists who will be using CDEP in their research.
>
> [1] https://arxiv.org/pdf/1703.03717.pdf
> [2] https://www.aclweb.org/anthology/N07-1033.pdf
> [3] https://arxiv.org/abs/1905.13714
> [4] Slide 30: http://theory.stanford.edu/~ataly/Talks/berkeley_ig_talk_feb_2019.pdf (talk from co-creator of integrated gradients)

---

### Public Comment · ~Joseph_David_Janizek1 · 2019-10-08
**Computational Efficiency of Expected Gradients**

Hi, I’m one of the authors of the Attribution Priors (Expected Gradients) method. Thank you for the citation — it’s always exciting to see more work on this relatively new research area!

We noticed that you said EG has high runtime and memory requirements because we recommend 200 samples per example - but our paper actually recommends exactly the opposite! In fact, all of our image experiments (MNIST and ImageNet in the supplement), use exactly 1 sample per example during training, which corresponds to no additional memory requirements and roughly the same training speed as Ross et al. (2017). This works because a single sample is an unbiased estimator for the true value of EG! Thus, this process regularizes the true value in expectation over many training steps.

We also notice that for the color MNIST problem, you choose to penalize the magnitude of the individual EG attributions (using an L2 penalty). One benefit of our attribution priors framework is that many human-intuitive priors (such as “attributions should be similar across color channels”) can be directly encoded as a penalty on the EG attributions. We believe such task-specific priors can lead to greatly improved performance.

We would be eager to see further comparisons with our method, and hope this insight allows for a more computationally-manageable workload.

---

> ### Author Response · Authors · 2019-10-09
> **Penalizing variance in attributions for color channels  does not improve performance meaningfully**
>
> Thanks for your feedback on how to better customize your method to the color MNIST task. Based on your suggestion, we ran an experiment which penalized the variance between attribution of different color channels, yielding a new accuracy of 10.3%. Seeing as baseline (random) accuracy is 10% on this dataset, 10.3% is not a meaningful gain, especially relative to the 25.2% accuracy our method achieves. In fact, this solution occurs only at a high enough penalty rate that the training accuracy goes down to near random.
>
> We will report these numbers, including the computational comparison, in an updated manuscript, when allowed to do so (after reviews have been returned).

---

### Author Response · Authors · 2019-11-07
**Thank you to reviewers - updates to manuscript**

We would like to thank all reviewers for their time and effort. We have responded to their concerns below, and made the following changes to the manuscript as a result:

- We have added references to Zaidan 2007 and Strout 2019

- Per the comment from Joseph Janizek (author of the expected gradients paper), we updated the computational and accuracy results for expected gradients. While improved, it still fails to beat a random baseline

- Improved ColorMNIST results: Previous results on ColorMNIST were non-deterministic  (despite a set random seed) due to a strange cuDNN setting. While rerunning those experiments we discovered that increasing the regularization parameter improves the mean accuracy using CDEP to 31% (previously 25.5%).  We have updated the manuscript accordingly.

---

### Decision · Program_Chairs · 2019-12-19

**Decision:**

Reject

**Comment:**

The paper contains interesting ideas for giving simple explanations to a NN; however, the reviewers do not feel the contribution is sufficiently novel to merit acceptance.